# Osteoblasts and Fibroblasts Interaction with a Porcine Acellular Dermal Matrix Membrane

**DOI:** 10.3390/ijms24043649

**Published:** 2023-02-11

**Authors:** Pietro Felice, Emira D’Amico, Tania Vanessa Pierfelice, Morena Petrini, Carlo Barausse, Maryia Karaban, Antonio Barone, Giovanna Iezzi

**Affiliations:** 1Department of Biomedical and Neuromotor Sciences, University of Bologna, 40126 Bologna, Italy; 2Department of Medical, Oral and Biotechnological Sciences, University G. d’Annunzio of Chieti-Pescara, 66100 Chieti, Italy; 3Postgraduate School of Oral Surgery, University of Modena and Reggio Emilia, 41121 Modena, Italy; 4Unit of Oral Surgery and Implantology, University Hospitals of Geneva, University of Geneva, 1205 Geneva, Switzerland; 5Complex Unit of Stomatology and Oral Surgery, Department of Surgical, Medical, Molecular Pathologies and of the Critical Needs, School of Dentistry, University of Pisa, University-Hospital of Pisa, 56124 Pisa, Italy

**Keywords:** dermal matrix membrane, collagen, human gingival fibroblasts, human oral osteoblasts, ALP, *OCN*, *COL1*, *FN1*, *MMP8*

## Abstract

The use of collagen membranes has remained the gold standard in GTR/GBR. In this study, the features and the biological activities of an acellular porcine dermis collagen matrix membrane applicable during dental surgery were investigated, and also by applying hydration with NaCl. Thus, two tested membranes were distinguished, the H-Membrane and Membrane, compared to the control cell culture plastic. The characterization was performed by SEM and histological analyses. In contrast, the biocompatibility was investigated on HGF and HOB cells at 3, 7, and 14 days by MTT for proliferation study; by SEM and histology for cell interaction study; and by RT-PCR for function-related genes study. In HOBs seeded on membranes, mineralization functions by ALP assay and Alizarin Red staining were also investigated. Results indicated that the tested membranes, especially when hydrated, can promote the proliferation and attachment of cells at each time. Furthermore, membranes significantly increased ALP and mineralization activities in HOBs as well as the osteoblastic-related genes *ALP* and *OCN*. Similarly, membranes significantly increased ECM-related and *MMP8* gene expression in HGFs. In conclusion, the tested acellular porcine dermis collagen matrix membrane, mainly when it is hydrated, behaved as a suitable microenvironment for oral cells.

## 1. Introduction

During guided tissue regeneration (GTR) and guided bone regeneration (GBR), the use of a membrane permits stabilizing the coagulum to prevent the colonization of the site by rapidly proliferating epithelium cells. It thus promotes the growth of slower-growing cells capable of forming bone and maintaining the space for regeneration. The characteristics of the membranes should be clinical manageability, space-making ability, biocompatibility, cell-occlusion properties, integration by the host tissues, and adequate mechanical and physical properties [1].

However, porcine acellular dermal matrix membranes (PADMs) have been introduced in the market to increase the gingival thickness or to treat gingival recessions as an alternative to connective tissue grafts (CTGs) [2]. The procedures of soft tissue augmentation around dental implants are beneficial in the case of marginal bone loss; a recent review has shown a bidirectional relationship between bone and soft tissues that act as a “guard” [3]. Soft tissue augmentation procedures around dental implants are used to increase attached keratinized mucosa to permit oral hygiene procedures, but also to increase soft tissue volumes, which permit compensation for the collapse of the vestibular cortex, which is very common following the loss of teeth [4].

It has been shown that the combination of hard and soft tissue augmentation procedures did not influence the marginal bone level around the implants but was significant for marginal soft tissue recession [3].

PADMs are characterized by a three-dimensional acellular network of collagen types I and III and elastin, in which cellular components and antigens have been removed to avoid any risk of tissue rejection [5].

The advantages of using a heterologous membrane permit the avoidance of connective tissue sampling surgery, with the associated risks of bleeding, increased surgical times, increased infectious risks, increased difficulty of the clinical case, and post-operative discomfort [6]. Moreover, a hydrated membrane type seems to offer better characteristics, comparing the direct and indirect cytotoxicity of a porcine-dried acellular dermal matrix in vitro. Both are used for periodontal and peri-implant soft tissue regeneration, always with the indication of maintaining a vascular bed on which to insert these matrices, even with numerous layers to increase the volume of the tissues around the implants.

Some disadvantages of PADMs are related to their malleability and degradation times. High stiffness leads to a difficult adjustment in the defect. However, if it is without any support, it will not support tissues, and malleability will be reduced. Another disadvantage of PADMs is the possibility of exposure in the oral cavity and subsequent contamination by bacteria. The presence of a high-cross-linking structure of the membranes influences their stiffness and the necessary time to fully reabsorb the material; the higher this parameter, the higher the risk of exposure in the oral cavity, contamination by bacteria, and induction of clinical failure [7].

Many studies have shown that PADMs are able to promote the growth and proliferation of human gingival fibroblasts, osteoblasts, and endothelial cells and to facilitate the migration, proliferation, and adhesion of periodontal ligament cells and human oral fibroblasts [8,9]. Furthermore, many studies supported the hypothesis that hydrated PADMs promoted better cell adhesion and proliferation, as well as faster and earlier revascularization than dried PADMs [10,11]. Consequently, many PADMs have been introduced on the market for several medical applications: wound coverage, eyelid repair, breast reconstruction, facial paralysis correction, abdominal wall reconstruction, stress urinary incontinence treatment, and rhinoplasty.

However, the process of production and the postprocessing treatments of PADMs have a great influence on their final three-dimensional conformation, porosity, mechanical stiffness, biodegradation process, and handling by the surgeon, also influencing their biological properties. 

In particular, PADMs produced by different manufacturers could differ from each other, especially in the collagen cross-links, because during production, they could be subjected to different physical treatments, such as ionizing or ultraviolet light irradiation, dehydrothermal treatment, and chemical methods involving aldehydes and isocyanates [12].

The aim of this study was to investigate the topographical and biological properties of a novel acellular dermal matrix membrane to evaluate its use in periodontology for soft tissue augmentation. Moreover, based on the contact of the PADMs with bone tissues, the biological response of the oral osteoblasts was tested.

## 2. Results

### 2.1. Characterization of the Membrane

The SEM observations permitted an evaluation of the structure of the membrane which appeared smoothed in both groups at the magnifications of 390, 1000, and 3000× (Figure 1). When the membrane was hydrated with NaCl (H-Membrane group) the surface became more porous than the manufactured membrane (Membrane). The histological images showed a mesh structure in both conditions (Figure 2). The manufactured membrane exhibited a closer mesh structure than the H-Membrane. The presence of a dense network of collagen fibers was observed using polarized light (Figure 2C,D).

### 2.2. Biological Activity

#### 2.2.1. Cell Proliferation

The growth of the gingival fibroblasts (HGFs) and oral osteoblasts (HOBs) was evaluated by MTT assay at 3, 7, and 14 days, as shown in Figure 3. At 3 days, HGFs cultured on the H-Membrane and Membrane statistically increased with respect to CTRL (*p* < 0.001). Comparing the growth of cells on the H-Membrane and Membrane, there were similar results (Figure 3A). Although, at 7 days, a slight increase in cell viability was observed on the H-Membrane and Membrane with respect to CTRL, and the results were not significant. Comparison between the H-Membrane and Membrane conditions projects similar results at 7 days. However, a significant enhancement of HGF proliferation was shown at 14 days on the H-Membrane and Membrane compared to CTRL (*p* < 0.0001). In addition, the growth of HGFs cultured on the H-Membrane showed an increase compared to the Membrane.

HOBs cultured on the H-Membrane showed a statistically augmented proliferation rate with respect to CTRL (*p* < 0.0001) at 3, 7, and 14 days (Figure 3B). In particular, at 3 days, growth levels in the Membrane and CTRL groups were comparable, but they increased slightly at 7 days and significantly at 14 days. Furthermore, at 3 days, the comparison of the H-Membrane and Membrane conditions revealed a significant increase in HOB proliferation in the H-Membrane versus Membrane (*p* < 0.0001). The same trend was observed at 7 and 14 days.

#### 2.2.2. Cell Attachment and Cell Morphology

SEM evaluated cell adhesion at 3, 7, and 14 days (Figure 4 and Figure 5). At 3 days, the SEM observation showed that the cells were uniformly distributed on the membranes in both tested conditions. HGFs appeared elongated and spindle-shaped with cytoplasmic extensions and lamellipodia (Figure 4A–F). At 7 days, a close confluence was observed that fully occurred at 14 days of culture (Figure 4G–L,M–R). At 14 days, at magnifications of 390×, a layer of fibroblasts was shown, but the morphology of cells was not notable (Figure 4M–R). At magnifications of 3000×, the interaction among cells was observed (Figure 4E,F,K,L,Q,R). No morphology changes were observed when comparing the H-Membrane and Membrane groups.

Concerning HOBs, because of the large number of cells attached to the membranes, the detailed morphology of cells was not visible at three days with a magnification of 390× (Figure 5A–F). A layer of cells colonized the entire surface of the membranes at 7 and 14 days (Figure 5G–R). Just at 3000× of magnification, the interactions of HOBs through cellular extensions can be observed (Figure 5E,F,K,L).

#### 2.2.3. Cell Interaction with the Membrane

At 400× of magnification, HGFs appeared rather elongated and more numerous at 3 days than at 7 days (Figure 6A–D). HOBs appeared to be larger than HGFs. They assumed a star-shaped appearance at 3 days (Figure 6E–H). At 7 days, the osteoblasts seeded on the hydrated membrane were more numerous, and the cells appeared more elongated and aggregated than those on the Membrane. 

#### 2.2.4. ALP Activity

At 7 days of culture, the enzymatic activity of Alkaline Phosphatase (ALP) was significantly enhanced when HOBs were seeded both on H-Membranes and Membranes with respect to CTRL (*p* < 0.0001) (Figure 7). In detail, the comparison of the H-Membrane and Membrane groups revealed that the ALP levels were statistically higher in the H-Membrane than Membrane.

#### 2.2.5. Mineralization

Calcium mineralization of HOBs was evaluated by Alizarin Red staining and quantified by adding cetylpyridinium chloride (CPC) at 14 days of culture (Figure 8). More mineralized nodules were observed in osteoblasts seeded on the H-Membrane and Membrane compared to CTRL. Furthermore, brighter red-colored nodules were shown in H-Membrane and Membrane conditions with respect to CTRL. The intensity of red was similar between the H-Membrane and Membrane groups (Figure 8A). These qualitative results were confirmed by CPC. Calcium deposit levels were statistically increased in the H-Membrane and Membrane with respect to CTRL (*p* < 0.0001). The levels of the H-Membrane and Membrane appeared comparable (Figure 8B).

#### 2.2.6. Gene Expression in Gingival Fibroblasts and Osteoblasts

The effects of the membrane on gingival fibroblasts and osteoblasts were investigated at 3, 7, and 14-days post-seeding utilizing real-time PCR (Figure 9 and Figure 10). In particular, the expressions of collagen 1 (*COL1*), fibronectin 1 (*FN1*), metalloprotease 8 (MMP8) in HGFs and *ALP*, and osteocalcin (*OCN*) were specifically examined. At 3, 7, and 14 days in HGFs, *COL1* was statistically upregulated in the H-membrane and Membrane groups compared to CTRL. HGFs cultured on the H-Membrane and Membrane showed similar levels of expression of *COL1* at 3 and 7 days, while at 14 days they were statistically higher in the H-Membrane than in the Membrane (Figure 9A). Globally, *FN1* expression exhibited a similar trend of *COL1*. In detail, at 3 days, FN1 expression was augmented in the H-Membrane and Membrane compared to CTRL (*p* < 0.05), but comparable levels were observed between the H-Membrane and Membrane. In contrast, the Membrane group showed a slightly diminished expression with respect to the H-Membrane at 7 and 14 days (Figure 9B). Although *MMP8* expression was slightly higher in the H-membrane and Membrane conditions compared to CTRL, the difference was not statistically significant. Just after 14 days of culture, *MMP8* levels showed a significant enhancement (*p* < 0.05). At all timing points, comparable levels of MMP8 expression were observed in the H-Membrane and Membrane groups (Figure 9C).

In HOBs, Figure 10A showed statistically augmented levels of *ALP* expression in the H-Membrane and Membrane compared to CTRL at 3, 7, and 14 days. The highest expression was observed at 7 and 14 days. The comparison between the H-Membrane and Membrane pointed to more increased *ALP* levels in the H-Membrane than Membrane (Figure 10A). *OCN* mRNA levels were statistically upregulated in the H-Membrane group compared to CTRL at all timing points. Concerning the Membrane condition, *OCN* levels were higher in the H-Membrane group than in the Membrane group (Figure 10B).

## 3. Discussion

This study aimed to evaluate the characteristics and biocompatibility of an acellular porcine dermal collagen matrix membrane applicable during dental surgery. The physical characteristics of membranes, such as porosity, surface topography, chemical composition, and stiffness, can influence guided tissue and guided bone regeneration (GBR/GTR) [13]. Thus, the first set of experiments investigated the surface morphologic features of membranes by SEM and histological analyses with and without previous hydration. The surface of the H-membrane appeared more porous than the Membrane under an SEM microscope. The histological analyses showed a mesh structure in both tested membranes, and the presence of a dense network of collagen fibers was highlighted using polarized light. The dehydrated membrane exhibited a closer mesh structure than the hydrated one, and these differences are discussed later as potential reasons for the observed cellular differences. 

Given that this type of membrane can be used for soft tissue grafting in combination with GBR/GTR in this study, the biological properties of the membrane were evaluated on gingival fibroblasts and on oral osteoblasts isolated from human biopsies. The growth of HGFs and HOBs was evaluated at 3, 7, and 14 days. At each time, both HGFs and HOBs seeded on the H-Membrane had a proliferative rate higher than those seeded on the Membrane. This is in line with the study of Guarnieri and co-workers, who observed a higher proliferation rate of HPMSCs seeded on hydrated PADM compared to the non-hydrated membrane [10]. In detail, for HGFs, a statistically significant increase in cell growth (*p* < 0.0001) was observed at 3 and 14 days, while at 7 days, the cellular metabolic activity was reduced. This observed trend may be characteristic of fibroblast cells that are usually critical components of wound healing, during which fibroblast accumulation begins 3–5 days after injury and may last up to 14 days, and it may have an up-and-down trend [14,15]. It was reported that after biomaterial implantation, fibroblasts undergo a response known as “activation,” characterized by a transition of quiescent cells into a myofibroblast-like phenotype. Responses of fibroblasts to activation include proliferation, fibrinogenesis, and the release of cytokine and proteolytic enzymes. These phases imply an up-and-down trend in the proliferation to permit the cells to move in the following phase of their biological response [15], whereas the proliferation rate of osteoblastic cells is usually more constant [16]. In this study, the proliferation of HOBs seeded on the H-membrane was significantly (*p* < 0.0001) increased with respect to the control in a time-dependent manner. Thereafter, the cell behavior on membranes was investigated for cell attachment, cell morphology, and cell interaction with the membrane’s surface through SEM and histological analyses. Notable differences emerged from SEM images between the two tested membranes, and both showed high biocompatibility to HGFs and HOBs that appeared strongly attached to the surface with typical morphology. One potential explanation may be that both membranes possess a dense network of collagen fibers, thereby providing a more easily attachable surface for incoming HGFs and HOBs. In addition, adhesion could be promoted through Cadherin activity [17,18]. Histological images, on the other hand, revealed an enhanced HOB interaction with the H-membrane compared to the Membrane without hydration. For this reason, for HOBs, the ALP activity as the main osteoblastic marker and mineralization activity were also investigated. The tested membranes significantly stimulated ALP’s enzymatic activity compared to the control (*p* < 0.0001), with the highest value for HOBs cultured on the H-Membrane. In a similar manner, the qualitative evaluation of calcium deposition by Alizarin Red staining at 14 days showed a more intense red color for both membranes compared to the control, indicating the presence of more mineralized nodules. Furthermore, the quantitative analysis by CPC revealed that calcium deposits resulted in the highest levels in the H-membrane. Increased mineralization rate and bone formation after surgery are fundamental, and here both membranes, especially the H-Membrane, seemed tp favor adhesion, proliferation, and mineralization activity in HOBs. Thus, the osteoblastic-related genes *ALP* and *OCN* were also investigated at the early and late stages of HOBs. *ALP* levels commonly indicate new bone formation and osteoblastic activity [19]. In this study, both membranes significantly stimulated the *ALP* mRNA levels with respect to the control, and the increase was maximum at the late stage of cells corresponding to 14 days post-seeding, with the H-Membrane demonstrating the highest results (*p* < 0.001) significantly. Osteocalcin is known to be a bone tissue-specific protein, and its expression is usually correlated to the expression of *ALP* [20]. As expected, the *OCN* gene expression was similar to the trend of *ALP* expression and both of these bone-related genes were up-regulated more in the H-Membrane compared to the Membrane at each time. Thereafter, ECM-related genes, including collagen type I α1 (*COL1*) and fibronectin (*FN1*), in human gingival fibroblast were also evaluated by real-time PCR. It was found that both membranes provoked a significant increase in *COL1* and *FN1* mRNA levels when compared to the control. *MMP8* (collagenase-2) belongs to the matrix metalloproteinase family of extracellular matrix-degrading enzymes, involved in tissue remodeling and wound repair. In particular, MMP8 has a protective role during the inflammatory response to various agents, despite the abnormal levels of this gene indicating chronic periodontitis [21]. Polymorphonuclear neutrophils are the main source of MMP8, but also, fibroblasts have been shown to produce it [22]. MMP8 has been shown to be the predominant collagenase in healing skin wounds, with a 100-fold increase of MMP8 in chronic wounds compared with acute wounds [23]. A previous study with MMP8-deficient (Mmp8−/−) mice reported delayed skin wound healing with altered neutrophil influx and an increase in TGF-β1, signaling molecule phosphorylated Smad (PSmad)-2 levels [24]. However, another study reported impaired rat skin wound healing in excess MMP-8 expression [25]. The partly contradictory findings demonstrate the complex role of MMP8 in tissue repair. Another study demonstrated the protective role of MMP8 in MMP8-deficiency mice; the authors observed a more severe alveolar bone loss in the *P. gingivalis* infection model [26]. Here, the expression of the *MMP8* gene in HGFs seeded on tested membranes was comparable to the control at 3 days of culture. It slightly increased at 7 days and it resulted significantly (*p* < 0.05) higher at 14 days of culture. The up-regulation of *COL1*, *FN1*, and *MMP8* may indicate that the tested membranes, especially the H-Membrane, are able to stimulate the renewal of the ECM. In addition, the level of MMP8 may be considered an input for fibroblasts involved in the wound repair process and for HOBs lined on the surface of membranes to appose bone matrix. Altogether, the results of this study suggested the acellular porcine dermal collagen matrix membrane has the capability to favor certain biological activities of the key cells in the oral cavity. The results of this study were in line with another study that evaluated the biological properties in vitro of PADMS on gingival fibroblasts and osteoblast-like cells [5]. However, the seeded cells were isolated from oral tissue, allowing the testing of PADM directly with cells of the oral cavity. Although most PADMs are produced and tested for gingival augmentation procedures, recent attempts have aimed at combining collagen membranes with various molecules or growth factors to speed up the healing process and improve the quality of regenerated hard tissues [27]. Fujioka-Kobayashi and co-workers recently showed that PADMs combined with rhBMP-9 accelerated the osteopromotive potential of ST2 cells. Other common additives in in vitro experiments were Sr-CaP nanoparticles [28] and octacalcium phosphate (OCP) [29]. However, the combination of PADMs with additional substance requires further manipulation and technique to be applied, and whether these modified membranes would be effective for a long time in clinical settings remains to be seen. Here, the tested membranes, especially hydrated membranes, demonstrated an ability in stimulating the main activities of HGFs and HOBs, without any additional molecules [30,31,32]. Probably, the hydration condition may produce a membrane that is more similar to the in vivo environment suitable for oral cells.

Many studies demonstrated the correlation between surface features and the behavior of cells [33,34,35]. A recent review provides insight into the importance of membrane pores in controlling cellular responses [36]. Here, SEM and histological analyses revealed a porous structure that seems to favor the biological activities of both HGF and HOB, demonstrated by the higher expression of functional genes, above all the genes related to ECM components. These results are in accordance with a recent study that observed an improvement of the adhesion and the proliferation of chondrocytes seeded on a high porous scaffold with respect to a low porous one [37]. The present in vitro study aimed to primarily evaluate the main biological characteristics and potential of a novel PADM to better understand the clinical performance we can expect from this product. The clinical indication from this first study is to hydrate this specific type of resorbable membrane before its surgical placement to have a more favorable regenerative environment. Further clinical studies will be conducted in order to assess its clinical performance compared to previous study results. The promising findings of this study will drive us to further in vitro and in vivo studies.

## 4. Materials and Methods

### 4.1. Experimental Design

The dermal matrix was first characterized, and the biological activities of human gingival fibroblasts (HGFs) and human oral osteoblasts (HOBs) were then assessed. The following experimental conditions were used:i.CTRL: cells seeded on the plate;ii.H-Membrane: cells cultured on the matrix hydrated three times with NaCl 0.9%;iii.Membrane: cells seeded on the membrane used as manufactured.

Firstly, the adhesion and the proliferation of HGFs and HOBs cultured on the matrix were assessed at 3, 7, and 14 days. In particular, the Phosphatase Alkaline (ALP) levels of HOBs at 7 days; the mineralization of HOBs at 14 days; and the gene expression of constitutive markers of HGFs and HOBs at 3, 7, and 14 days were performed in order to evaluate the activity of cells on the matrix. 

### 4.2. Characterization of the Material

The membrane (Cellis Dental), produced by Cellis Dental (La Rochelle, France) was a non-pyrogenic collagen type I&III matrix obtained from porcine skin. It was supplied dry, sterilized, and did not contain any preservatives. In in vitro tests, the matrix was cut into squares of 5 mm × 5 mm under sterilized conditions.

#### 4.2.1. SEM Analyses

After rinsing with PBS and fixing for 1 h with 2.5% of glutaraldehyde at room temperature, the membranes were washed with PBS. Then, they were dehydrated in increasing concentrations of alcohol (from 40 to 100%) for 10 min each. The specimens were sputter coated with gold and examined using SEM (Philips XL20; Philips Inc., Eindhoven, the Netherlands). The images were taken at 390, 1000, and 3000×.

#### 4.2.2. Histological Analysis

The specimens were fixed in 10% buffered formalin and dehydrated in an ascending alcohol series. They were embedded in a glycol methacrylate resin (Technovit 7200 VLC; Kulzer, Wehrheim, Germany) and then polymerized. Sections of about 150 μm were obtained following the long axis of the devices. The sections were subsequently ground down to about 30 μm in width and stained with fuchsin and toluidine blue. The images were taken by an optical microscope (Leica, Wetzlar, Germany) at 40× and with polarized light.

### 4.3. Biological In Vitro Tests

#### 4.3.1. Cell Isolation

Human oral osteoblasts (HOBs) were derived from bone chips of n° 12 patients undergoing the surgical removal of lower third molars at the dental clinic of the G. D’Annunzio University according to Ethical Committee reference numbers: BONEISTO N. 22-10.07.2021. The culture procedure was conducted according to Pierfelice TV et al., 2022 [38]. Briefly, bone fragments were subjected to three enzymatic digestions at 37 °C for 20, 40, and 60 min using collagenase type 1A (Sigma-Aldrich, St. Louis, MO, USA) and trypsin-EDTA 0.25% (Corning, New York, NY, USA). After each digestion, this solution was centrifuged at 1200 rpm for 10 min and the pellet obtained was transferred into a T25 culture flask with low-glucose (1 g/L) DMEM supplemented with 10% FBS (SIAL, Rome, Italy), 1% antibiotics (100 µg/mL-1 streptomycin and 100 IU/mL-1 penicillin), and 1% L-glutamine (Corning) at 5% CO_2_ and 37 °C. The medium was changed every 4–5 days. 

#### 4.3.2. Cell Culture

Primary human gingival fibroblasts (HGF) were purchased from ATCC (Manassas, VA, USA) and were cultured in DMEM low glucose (Corning), supplemented with 10% fetal bovine serum (FBS) (SIAL), 1% penicillin, and streptomycin (Corning) at 37 °C and 5% CO_2_. Both cell lines, HOBs, and HGFs, were used from the 3rd and 5th passages. They were seeded on the top of the matrix at different densities and timing points depending on the test.

#### 4.3.3. Cell Viability

To assess the number of vital cells in each group, 3-(4,5-dimethylthiazol-2-yl)-2,5-diphenyl tetrazolium bromide (MTT) test was performed at 3, 7, and 14 days after the seeding of 104 cells/membrane using MTT (Sigma Aldrich, St. Louis, MO, USA) according to the manufacturer’s instructions. 

The absorbance was determined by a microplate reader (Synergy H1 Hybrid BioTek Instruments, Winooski, VT, USA) at 650 nm wavelength. The results were expressed as percentages and were calculated with respect to control (CTRL).

#### 4.3.4. Cell Adhesion

To observe cell attachment, scanning electron microscope (SEM) images were taken 3, 7, and 14 days after cell seeding (10^4^ cells/membrane). First, cells were fixed by 2.5% glutaraldehyde for 1 h followed by dehydration using sequential concentrations of ethanol. Then, they were sputtered with gold and observed at 390×, 1000×, and 3000× under an SEM (Philips XL20; Philips Inc., Eindhoven, the Netherlands) at 15 kV. 

#### 4.3.5. Cell Interaction with the Membrane

Cells at the density of 10^4^ were cultured on the membrane. The interaction of cells with the matrix was evaluated by histological analysis using blue di toluidine and fuchsine staining, as described in paragraph 4.2.2. The images were taken at the magnification of 400×.

#### 4.3.6. ALP Assay

HOBs at the density of 5 × 10^4^ were seeded on the top of the matrix for 7 days. The ALP assay kit, colorimetric AB83369 (Abcam Inc., Cambridge, UK), was used to assess ALP levels in accordance with the manufacturer’s instructions and as reported by D’Amico E et al., 2022 [39].

#### 4.3.7. Alizarin Red Staining and Quantification of Calcium Deposition

To evaluate the deposition of calcium nodules, 5 × 10^4^ HOBs/membrane were seeded on the membrane for 14 days. The qualitative analysis was performed using Alizarin Red Staining (ARS) (Sigma Aldrich, St. Louis, MO, USA) and the quantization using a 10% Cetylpyridinium Chloride (CPC) solution (Sigma-Aldrich, St. Louis, MO, USA). These analyses were carried out in accordance with the protocol outlined by Pierfelice TV et al. 2022 [40].

#### 4.3.8. Gene Expression

RT-PCR was performed with the objective of evaluating the gene expression of different markers in a certain stage of osteoblastic cells (ALP and OCN) and constitutive markers of fibroblast activity (COL1, FN1, MMP8). Total RNA was isolated using the Trifast reagent (EuroClone, Pero (MI), Italy). RNA was quantified on a Nanophotometer NP80 spectrophotometer (Implen NanoPhotometer, Westlake Village, CA, USA) for analysis of RNA integrity, purity, and concentration. Afterward, the GoTaq^®^2 Step RT-qPCR Kit (Promega, Madison, WI, USA) was used to obtain complementary DNA (cDNA) according to the manufacturer. SYBR Green (GoTaq^®^ 2 Step RT-qPCR Kit, Promega) was used to perform RT-qPCR. An amount of 10 µL of mixes, composed of 1 µL of cDNA, 0.2 µL of primer mixture, and 5 µL of master mix, was plated in a 96-well plate, and gene expression was determined using Quant Studio 7 Pro Real-Time PCR System (ThermoFisher, Waltham, MA, USA). The results were normalized to β-actin (β-ACT) for HOB and to Glyceraldehyde-3-Phosphate Dehydrogenase (GAPDH) using the 2 ^−ΔΔCt^ method. Primer sequences are reported in Table 1.

#### 4.3.9. Statistical Analysis

One-way ANOVA test followed by Tukey post hoc test was applied for the comparison of values among different groups at different time points. Statistical analysis was performed using GraphPad Prism8 (GraphPad Software, San Diego, CA, USA) with a significance level of 0.05.

## 5. Conclusions

In conclusion, this study demonstrated that the tested PADMs, when hydrated, behaved as a suitable microenvironment for gingival fibroblasts and oral osteoblasts with significantly superior soft tissue and osteopromotive properties compared with the control. Nevertheless, these studies point to the necessity of further evaluation for both membranes in vivo.

## Figures and Tables

**Figure 1 ijms-24-03649-f001:**
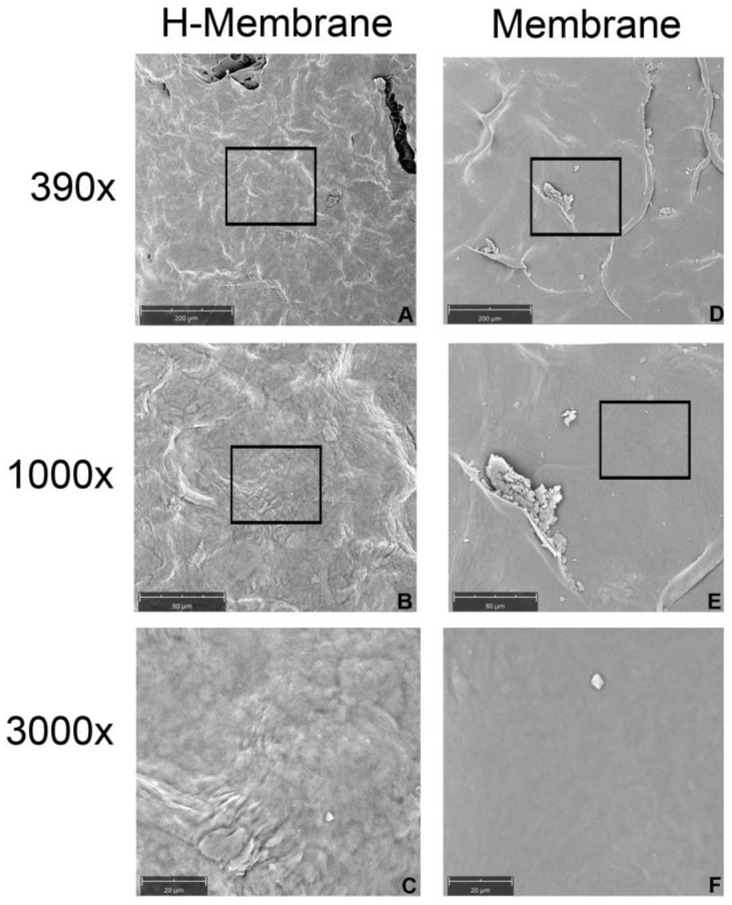
Scanning electron micrograph (SEM) images of the membranes. (**A**–**C**) hydrated membranes (H-Membrane); (**D**,**F**) manufactured membrane (Membrane); Mag= 390× (**A**,**D**), Magnification = 1000× (**B**,**E**), Magnification = 3000× (**C**,**F**). The squares indicate the magnified area.

**Figure 2 ijms-24-03649-f002:**
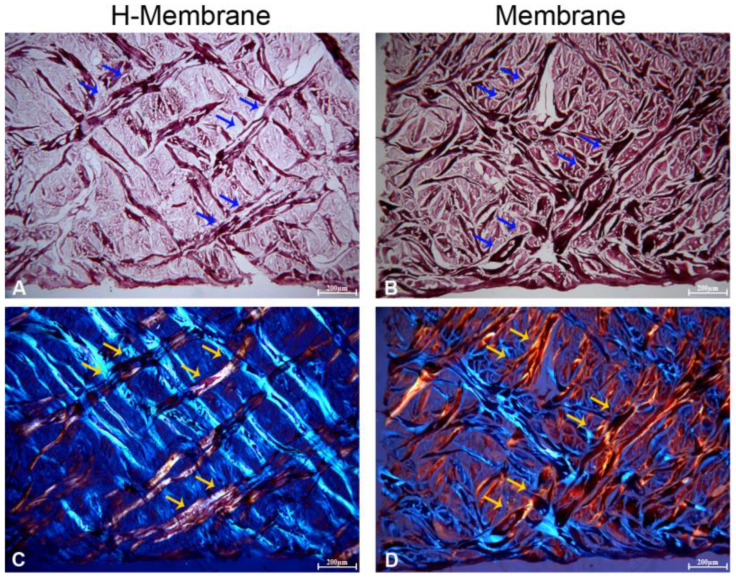
Characterization of the structure of the membrane by histological analysis. Hydrated membrane (H-Membrane) (**A**,**C**); Manufactured membrane (Membrane) (**B**,**D**), Magnification = 40×. Blue and yellow arrows indicate the collagen fibers.

**Figure 3 ijms-24-03649-f003:**
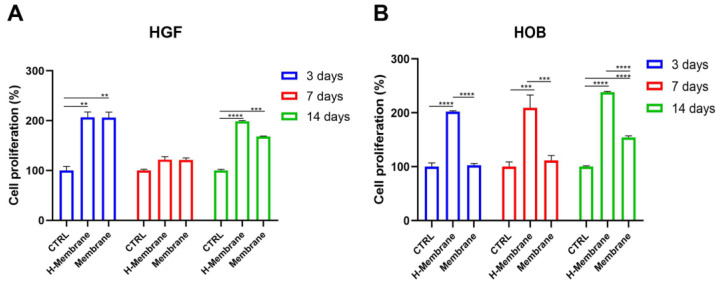
Cell proliferation of H-Membrane and Membrane after 3, 7, and 14 days of HGFs (**A**) and hOBs (**B**) cultures. (** *p*<0.001; ***, **** *p* < 0.0001).

**Figure 4 ijms-24-03649-f004:**
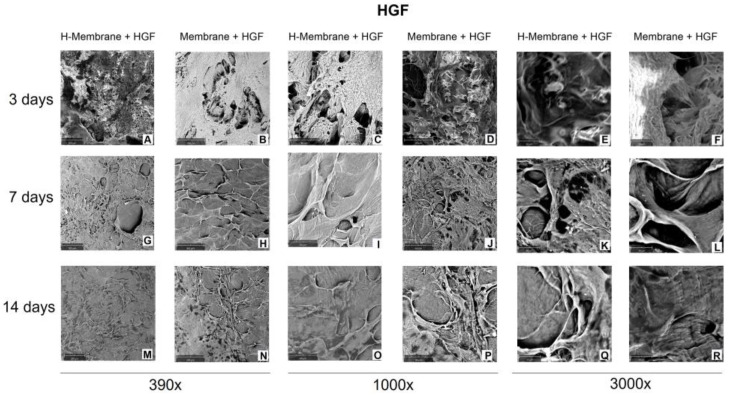
Scanning electron micrograph (SEM) images of HGFs seeded on H-Membrane and Membrane at 3 (**A**–**F**), 7 (**G**–**L**), and 14 (**M**–**R**) days. (Magnification = 390, 1000, and 3000×).

**Figure 5 ijms-24-03649-f005:**
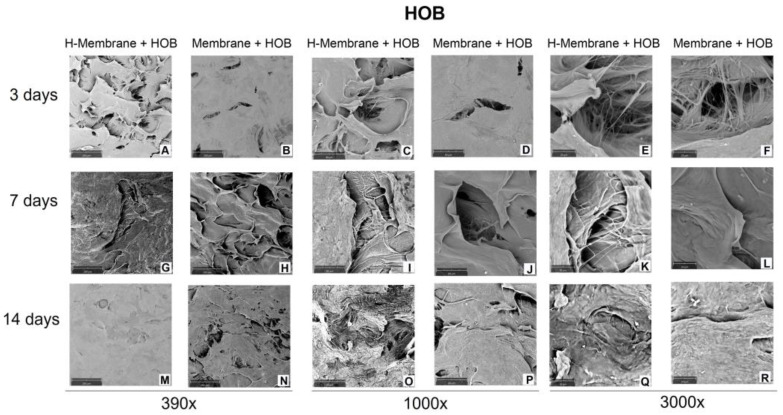
Scanning electron micrograph (SEM) images of HOBs cultured on H-Membrane and Membrane at 3 (**A**–**F**), 7 (**G**–**L**), and 14 (**M**–**R**) days. (Magnification = 390, 1000, and 3000×).

**Figure 6 ijms-24-03649-f006:**
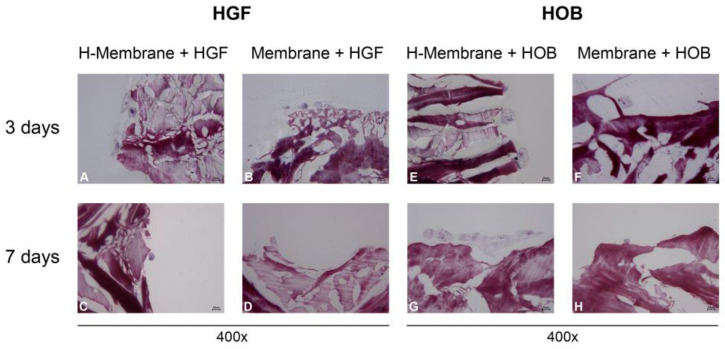
Cell interaction of HGFs (**A**–**D**) and HOBs (**E**–**H**) with H-Membrane and Membrane at 3 and 7 days. (Magnification = 400×).

**Figure 7 ijms-24-03649-f007:**
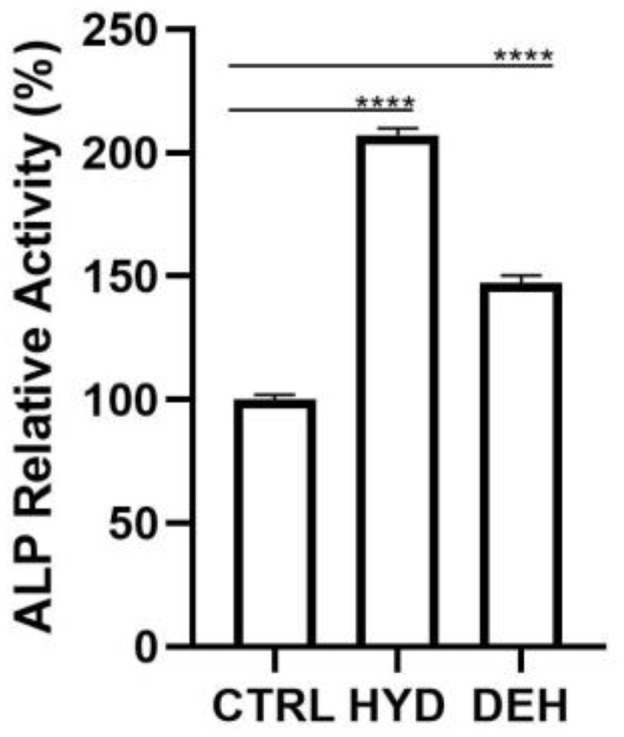
ALP activity of osteoblasts cultured for 7 days (**** *p* < 0.0001).

**Figure 8 ijms-24-03649-f008:**
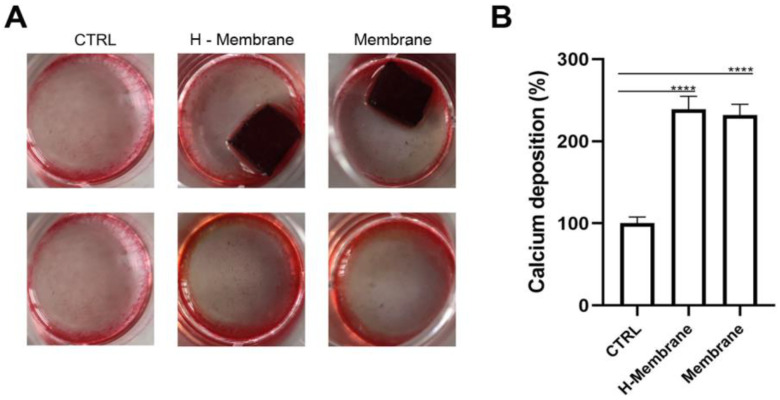
Alizarin red staining on osteoblasts cultured for 14 days (**A**), calcium deposition quantification at 14 days (**B**), (**** *p* < 0.0001).

**Figure 9 ijms-24-03649-f009:**
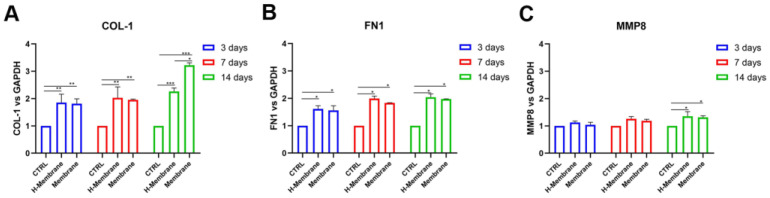
Real-time PCR of fibroblasts (HGFs) seeded on H-Membrane and Membrane for genes encoding Collagen 1 (COL1) (**A**), Fibronectin 1 (FN1) (**B**), Metalloprotease 8 (**C**) at 3, 7, and 14 days post-seeding (* *p* < 0.05; ** *p* < 0.001; *** *p* < 0.0001). The expression of COL-1, FN1, and MMP8 were upregulated in H-Membrane and Membrane with respect to CTRL.

**Figure 10 ijms-24-03649-f010:**
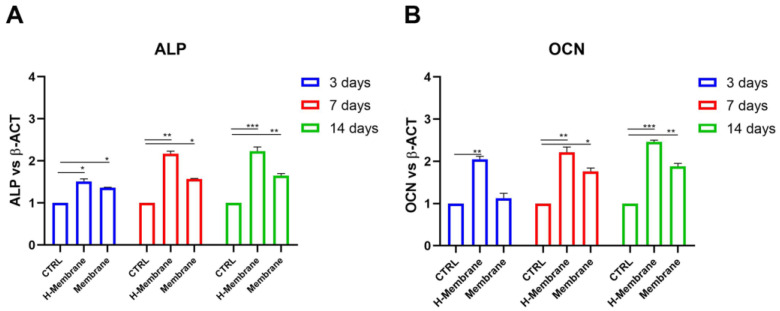
Real-time PCR of osteoblasts (HOBs) seeded on H-Membrane and Membrane for genes encoding Alkaline Phosphatase (ALP) (**A**) and Osteocalcin (OCN) (**B**) at 3, 7, and 14 days post-seeding (* *p* < 0.05; ** *p* < 0.001; *** *p* < 0.0001). ALP and OCN were more expressed in H-Membrane than in Membrane.

**Table 1 ijms-24-03649-t001:** Primer sequences used in RT-qPCR.

Gene	Forward Primer (5′–3′)	Reverse Primer (5′–3′)
*OCN*	TCAGCCAACTCGTCACAGTC	GGCGCTACCTGTATCAATGG
*ALP*	AATGAGTGAGTGACCATCCTGG	GCACCCCAAGACCTGCTTTAT
*COL1*	AGTCAGAGTGAGGACAGTGAATTG	CACATCACACCAGGAAGTGC
*FN1*	GGAAAGTGTCCCTATCTCTGATACC	AATGTTGGTGAATCGCAGGT
*MMP8*	ATGTTCTCCCTGAAGACGCT	AGACTGATACTGGTTGCTTGGT
*Β-ACT*	CCAGAGGCGTACAGGGATAG	GAGAAGATGACCCAGGACTCTC
*GAPDH*	ACGGGAAGCTTGTCATCAAT	GGAGGGATCTCGCATTTCTT

## Data Availability

The data that support the findings of this study are available from the corresponding author upon reasonable request.

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
