# Peer review of "Osteoblasts and Fibroblasts Interaction with a Porcine Acellular Dermal Matrix Membrane"

_ijms, 2023, doi:10.3390/ijms24043649_

Round 1
Reviewer 1 Report
Dear authors,
congratulations for the interesting topic addressed. To facilitate the understanding of some details, please consider the following:
-please specified some risks of using PADMs, to bring arguments in comparison with other methods 59-62
-Figure 1, 2-can the areas of interest be indicated to be more easily observed?
- 281-290 please compare the results with other studies
- consider some spelling errors
Author Response
#1
Dear authors,
congratulations for the interesting topic addressed. To facilitate the understanding of some
details, please consider the following:
-please specified some risks of using PADMs, to bring arguments in comparison with other
methods 59-62
AUTHOR’S ANSWER: Thank you for your comment. Some disadvantages of PADMs are
related to the malleability, and to the degradation-times. If the membrane is too rigid, it is
difficult for the surgeon to adapt to the defect, however, if it is without any sustain, it will not
sustain the tissues and the malleability will be reduced. Another disadvantage of PADMs is
the possibility to expose in the oral cavity, and consequently to be contaminated by bacteria.
The presence of high-cross linking structure of the membranes influences their stiffness and
also the necessary time to full reabsorb the material: higher is this parameter, higher is the
risk to expose in the oral cavity, to contaminate with bacteria, and to induce a clinical failure.
So, the ideal material, should promote the cellular interaction, proliferation, and
vascularization, in order to increase the healing process and being reabsorbed in a shorter
period [Barbeck M, Lorenz J, Kubesch A, Böhm N, Booms P, Choukroun J, Sader R,
Kirkpatrick CJ, Ghanaati S. Porcine Dermis-Derived Collagen Membranes Induce
Implantation Bed Vascularization Via Multinucleated Giant Cells: A Physiological Reaction?
J Oral Implantol. 2015 Dec;41(6):e238-51. doi: 10.1563/aaid-joi-D-14-00274. Epub 2014
Dec 29. PMID: 25546240.].
-Figure 1, 2-can the areas of interest be indicated to be more easily observed?
AUTHOR’S ANSWER: Thank you for your comment, we agree with you. We have
introduced squares in the Fig.1 and arrows in the Fig.2 to indicate the area of interest. In
detail, in Fig. 1 the squares underline the magnified area, while in Fig. 2 the mesh structure
is highlighted. Furthermore, in Fig. 1 there was an error with the magnification that we have
corrected.
- 281-290 please compare the results with other studies
AUTHOR’S ANSWER: Thank you for your comment. As your suggestion we added more
studies.
The results of this study were in line with another study that evaluated in vitro the biological
properties of PADMS on gingival fibroblasts and osteoblasts-like cells [doi:
10.1111/jre.12115]. Although, the majority of PADMs are produced and tested for gingival
augmentation procedures, recent attempts have aimed at combining collagen membranes
with various molecules or growth factors to speed up the healing process and improve the
quality of regenerated hard tissues [doi: 10.3389/fbioe.2022.921576.]. Fujioka-Kobayashi
and co-workers, recently showed that PADMs combined with rhBMP-9 accelerated the
osteopromotive potential of ST2 cells [doi: 10.11607/jomi.5652]. Other common additives in
vitro experiments were Sr-CaP nanoparticles [doi: 10.1080/09205063.2019.1646628],
octacalcium phosphate (OCP) [10.1016/j.msec.2018.12.115]. However, the combination of
PADMs with additional substance requires further manipulation and technique to applied,
and whether these modified membranes would be effective for long time, in clinical settings,
remains to be seen. Here the tested membranes, especially hydrated membrane
demonstrated ability in stimulating the main activities of HGFs and HOBs, without any
additional molecules on the surface.
- consider some spelling errors
AUTHOR’S ANSWER: Thank you for your comment. We have checked the English.
Reviewer 2 Report
Dear Authors,
you made a really great work!
However, some improvements are mandatory before acceptance.

Author Response
#2
The paper is an original research article on the osteoblasts and fibroblasts interaction with
a novel porcine acellular dermal matrix membrane.
The Authors made a great work in terms of methodology and the paper sounds scientific
and well written.
AUTHOR’S ANSWER: Thank you for your comment.
However, some improvements are mandatory before acceptance.
The abstract is well written, complete and summary in its various aspects. The keywords
are complete and appropriate. Please separate the keywords "Alp OCN Col1 FN1 MMP8".
AUTHOR’S ANSWER: Thank you for your kind suggestion. We have separated the
keywords.
In the introduction:
• “The advantages of the use of a heterologous membrane, permit to avoid the surgery of
connective tissue sampling, with the associated risks of bleeding, increased surgical times,
increased infectious risks, increased difficulty of the clinical case, and post-operative
discomfort[6].” I suggest to the authors to enrich the introduction underlining how a
membrane of this hydrated type seems to offer better characteristics, comparing a direct
and indirect cytotoxicity of a porcine dried acellular dermal matrix (PDADM) versus a
porcine hydrated acellular dermal matrix (PHADM) in vitro. Both are used for periodontal
and peri-implant soft tissue regeneration, always with the indication of maintaining a
vascular bed on which to insert these matrices, even with numerous layers to increase, as
well underlined by the authors, the volume of the tissues around the implants, as underlined
by:
“Guarnieri R, Reda R, Di Nardo D, Miccoli G, Zanza A, Testarelli L. In Vitro Direct and
Indirect Cytotoxicity Comparative Analysis of One Pre-Hydrated versus One Dried Acellular
Porcine Dermal Matrix. Materials (Basel). 2022 Mar 5;15(5):1937. doi:
10.3390/ma15051937.”
AUTHOR’S ANSWER: Thank you for your observation, and for suggesting this very
interesting paper. We have emphasized the use of hydrated membrane in the introduction.
The following sentence was added:
“Furthermore, many studies supported the hypothesis that the hydrate PADM compared to
the dried ones form promoted a better cell adhesion and proliferation and a faster and earlier
revascularization [“Guarnieri R, Reda R, Di Nardo D, Miccoli G, Zanza A, Testarelli L. In
Vitro Direct and Indirect Cytotoxicity Comparative Analysis of One Pre-Hydrated versus One
Dried Acellular Porcine Dermal Matrix. Materials (Basel). 2022 Mar 5;15(5):1937.
doi:10.3390/ma15051937; ”Nica, C.; Lin, Z.; Sculean, A.; Aspa-ruhova, M.B. Adsorption and
Release of Growth Factors from Four Different Por-cine-Derived Collagen Matrices.
Materials 2020, 13, 2635; Soundararajan, M.; Kannan, S. Fibroblasts and mesenchymal
stem cells: Two sides of the same coin? J. Cell Physiol. 2018, 233, 9099–9109]”.
Furthermore, we also reconsidered this concept in the discussion, as reported below: “This
is in line with the study of Guarnieri and coworkers, that observed the higher proliferation
rate of HPMSCs seeded on hydrated PADM compared non-hydrated membrane.”
I suggest to the authors to immediately improve the character of the captions following the
indications of the Journal to streamline the subsequent editing phases.
AUTHOR’S ANSWER: Thank you for your observation. We have standardized the caption
of the figures according to the journal guidelines, that suggest the use of a 9 pt.
Results are easy to understand and comprehensive. All the studied characteristics were
reported in tables which are clear and concise.
AUTHOR’S ANSWER: Thank you for your comment.
Discussion: this section is complete and evaluates the outcome of different papers present
in literature. The overall is comprehensive, concise and complete in its various aspects.
In the discussion:
• Considering the inflammatory pathways activated by MMP-8, with reference to "The MMP8
is a member of the matrix metalloproteinase family of extracellular matrix-degrading
enzymes that are involved in tissue remodeling and wound repair. In particular MMP8 has
a protective role during the inflammatory response to varoius agents, despite the abnormal
levels of this gene indicates chronic periodontitis[13]." How do you evaluate the use of
these membranes at the peri-implant level? Do you think they can modify the already rather
high values of inflammatory mediators at the peri-implant level?
AUTHOR’S ANSWER: Thank you for your observations. PADMs are already used to
increase the tissues thickness around implants, showing encouraging results. Indeed,
literature has shown that Peri-implant Keratinized mucosa width <2 mm was associated with
increased mean plaque index (mpi) and more discomfort after toothbrushing, consequently
also the inflammatory status of peri-implant tissues could be affected. [The role of
keratinized mucosa width as a risk factor for peri‐implant disease: A systematic review,
meta‐analysis, and trial sequential analysis, Andrea Ravidà].
The advantage of using PADMs respect connective tissue grafts is the absence of the donor
site morbidity. However, a recent review concluded that more data are necessary to confirm
that PADM can be considered as a suitable alternative to connective tissue graft [Papi P,
Pranno N, Di Murro B, Pompa G. Early implant placement and peri-implant augmentation
with a porcine-derived acellular dermal matrix and synthetic bone in the aesthetic area: a 2-
year follow-up prospective cohort study. Int J Oral Maxillofac Surg. 2021 Feb;50(2):258-266.
doi: 10.1016/j.ijom.2020.07.002. Epub 2020 Jul 14. PMID: 32680808; Dadlani S. Porcine
Acellular Dermal Matrix: An Alternative to Connective Tissue Graft-A Narrative Review. Int
J Dent. 2021 Sep 6;2021:1652032. doi: 10.1155/2021/1652032. PMID: 34527053; PMCID:
PMC8437668]. Furthermore, the peri-implant soft tissue phenotype (PSP) encompasses the
keratinized mucosa width (KMW), mucosal thickness (MT), and supracrestal tissue height
(STH). A recent review of Tavelli et al. has shown that the bilaminar approach on periimplant tissues involving connective tissue graft (CTG) or acellular dermal matrix (ADM)
obtained the highest amount of MT gain. In contrast, apically positioned flap (APF) in
combination with free gingival graft (FGG) was the most effective technique for increasing
KMW. KMW augmentation via APF was associated with a significant reduction in probing
depth, soft tissue dehiscence and plaque index, regardless of the soft tissue grafting material
employed, whereas bilaminar techniques with CTG or collagen matrix (CM) showed
beneficial effects on marginal bone level stability [Tavelli L, Barootchi S, Avila-Ortiz G, Urban
IA, Giannobile WV, Wang HL. Peri-implant soft tissue phenotype modification and its impact
on peri-implant health: A systematic review and network meta-analysis. J Periodontol. 2021
Jan;92(1):21-44. doi: 10.1002/JPER.19-0716. Epub 2020 Aug 9. PMID: 32710810].
Considering the role of MMP-8 in osteoclastogenesis.
AUTHOR’S ANSWER: thanks for your insightful questions.
Polymorphonuclear neutrophils are the main source of MMP8, but also fibroblasts have
been shown to produce it [R. Hanemaaijer, et al., 1997]. MMP-8 has been shown to be the
predominant collagenase in healing skin wounds, with a 100-fold increase of MMP-8 in
chronic wounds compared with acute wounds [ B.C. Nwomeh, et al., 1999]. A previous study
with MMP-8-deficient (Mmp8-/-) mice reported delayed skin wound healing with altered
neutrophil influx and increase in TGF-β1 signaling molecule phosphorylated Smad
(PSmad)-2 levels [A. Gutiérrez-Fernández, et al., 2007]. However, another study reported
impaired rat skin wound healing in an excess of MMP-8 expression [P.L. Danielsen, et al.,
2011]. The partly contradicting findings demonstrate the complex role of MMP-8 in tissue
repair. Another study demonstrated the protective role of MMP8, in a MMP8-deficiency mice
the authors observed a more severe alveolar bone loss in P. gingivalis infection model
[Kuula, H.; Salo, T.; Pirila, E.; Tuomainen, A.M.; Jauhiainen, M.; Uitto, V.J.; Tjaderhane, L.;
Pussinen, P.J.; Sorsa, T. Local and systemic responses in matrix metalloproteinase 8-
deficient mice during Porphyromonas gingivalis-induced periodontitis. Infect. Immun. 2009,
77, 850–859.].
The expression of MMP8 seems to be correlated to another member of collagenases, such
as MMP13, in various cells in skeletal system, including osteoblasts and osteoclasts
[Andersen, T.L.; del Carmen Ovejero, M.; Kirkegaard, T.; Lenhard, T.; Foged, N.T.;
Delaisse, J.M. A scrutiny of matrix metalloproteinases in osteoclasts: Evidence for
heterogeneity and for the presence of MMPs synthesized by other cells. Bone 2004, 35,
1107–1119; Sasano, Y.; Zhu, J.X.; Tsubota, M.; Takahashi, I.; Onodera, K.; Mizoguchi, I.;
Kagayama, M. Gene expression of MMP8 and MMP13 during embryonic development of
bone and cartilage in the rat mandible and hind limb. J. Histochem. Cytochem. 2002, 50,
325–332;]. MMP8-deficiency led to an increased joint inflammation and bone loss in serum
transfer arthritis model [Garcia, S.; Forteza, J.; Lopez-Otin, C.; Gomez-Reino, J.J.;
Gonzalez, A.; Conde, C. Matrix metalloproteinase-8 deficiency increases joint inflammation
and bone erosion in the K/BxN serum-transfer arthritis model. Arthritis Res. Ther. 2010, 12,
R224.]. However, the expression and function of collagenases, i.e., MMP8, in osteoclasts
are yet to be thoroughly studied, for this reason we had not considered in our study, a
possible role of MMP8 in the osteoclastogenesis.
Materials and methods are clear and well explained. Different aspects are analyzed with a
dedicated statistical test. The authors did a great job in the explication of all the variables
identified and included in the study.
Conclusions are concise and clear.
AUTHOR’S ANSWER: Thank you for your comment.
Bibliography should be formatted respecting the journal’s requirements and no improper
citations are evidenced.
AUTHOR’S ANSWER: Thank you for your request. We have formatted the bibliography
according to the guideline.
Figures and labels are clear and easy to comprehend.
English is clear and easy to understand
AUTHOR’S ANSWER: Thank you for your comment.
Author Response
#3
Review manuscript: IJMS-2160763
This research article reports the osteoblasts and fibroblasts interaction with porcine acellular
dermal matrix membranes using Hydrated (H-membrane) and non-hydrated (Membrane)
membrane conditions. Research is well defined and well characterized, presented
appropriately throughout the manuscript. However, the novelty is less because the role of
osteoblast and fibroblast is reported for various applications and similarly the lack of in vivo
study results. My comments are as follows;
Major:
1. What is the novelty of this work? Role of osteoblast and fibroblasts in tissue modelling
and bone modelling. Some of the previous studies already determined the use of porcine
acellular dermal matrix membrane for gingival augmentation in periodontal surgeries and
are reported by performing both in vitro and in vivo characterizations. Example
Ref.doi.org/10.1111/jre.12115; doi: 10.1155/2018/6406051; doi: 10.1155/2021/1652032).
AUTHOR’S ANSWER: Thank you for your observation. This matrix can be considered novel
compared to the other PADMs because it is produced by a different manufacturer, and
consequently are subjected to different treatments: physical stimuli ionizing or ultravioletlight irradiation, dehydrothermal treatment, and chemical methods, such as aldehydes and
isocyanates. It also could be differed each other’s, especially in the collagen’s crosslinks.
These modifications could influence the biological response. These aspects are reported in
the introduction.
Moreover, concerning the novelty of the study, in this work is identified the potential role of
MMP8 in the wound healing and tissue regeneration. We have observed slightly increment
of MMP8 gene expression, that resulted a positive and stimulating factor of wound healing.
In contrast, high levels of MMP8 lead chronic inflammation and consequent tissue
degradation [Zhang L, Li X, Yan H, Huang L. Salivary matrix metalloproteinase (MMP)-8 as
a biomarker for periodontitis: A PRISMA-compliant systematic review and meta-analysis.
Medicine (Baltimore). 2018 Jan;97(3):e9642. doi: 10.1097/MD.0000000000009642. PMID:
29504999; PMCID: PMC5779768; Sorsa T, Tervahartiala T, Leppilahti J, Hernandez M,
Gamonal J, Tuomainen AM, Lauhio A, Pussinen PJ, Mäntylä P. Collagenase-2 (MMP-8) as
a point-of-care biomarker in periodontitis and cardiovascular diseases. Therapeutic
response to non-antimicrobial properties of tetracyclines. Pharmacol Res. 2011
Feb;63(2):108-13. doi: 10.1016/j.phrs.2010.10.005. Epub 2010 Oct 16. PMID: 20937384].
The slightly enhanced levels of MMP8, identified after the use of this matrix, could reduce
the risks observed with the use of other PADMs. In detail, one of the major risks connected
with the use of PADM was represented by the exposure of sites that led to superinfection
and chronic inflammation. Therefore, this could suggest that the slightly enhanced levels of
MMP8 may prevent this problem. However, this is only a suggestion and further in vitro and
vivo studies are needed to demonstrate this aspect.
Furthermore, another novel aspect is represented from the use of the histological technique
in an in vitro study since the histology is usually used for tissues in vivo studies. Using the
histology we have not only showed the adhesion and the interaction of the cells with the
membrane, but also the polarized light has allowed to observe the network and the direction
of collagen fibers.
Similarly, what is the rationale behind this particular study?
AUTHOR’S ANSWER: thank you very much for your comment. The rationale of this study
was the increasing the knowledge about the biological mechanism and the cellular events
that arise when osteoblast and fibroblasts meet these types of membranes. The second
rationale was to verify eventual presence of differences in the biological response with and
without hydration of the membrane. We have chosen these types of cells, because PADMs
are used to increase the gingival thickness or to treat gingival recessions, as an alternative
to connective tissue grafts, however PADMs are also into contact with bone tissue, thus we
have evaluated osteoblast response.
We have added this concept in the aim: “Moreover, based on the contact of the PADMs with
bone tissues the biological response of the oral osteoblasts was tested.”
The authors should have highlighted the advantage of using a Hydrated membrane than the
previous study results. On the other hand, could have mentioned the specific disadvantages
or potential limits of using porcine acellular matrix membrane from previous study results.
AUTHOR’S ANSWER: Thanks to your comments. As previously explained, it is the first time
that this particular porcine derma membrane (Cellis Dental, Mec Cellis Biotech) has been
evaluated in vitro. No other studies results, about this membrane have been published. It is
important to highlight this concept because the processing methods used to produce the
PADMs can have a great influence on the presence of cross-linking in the materials, and
consequently in their interactions with cells and tissues.
Some disadvantages of PADMs are related to the malleability, and to the degradation-times.
If the membrane is too rigid, it is difficult for the surgeon to adapt to the defect, however, if it
is without any sustain, it will not sustain the tissues and also the malleability will be reduced.
Another disadvantage of PADMs is the possibility to expose in the oral cavity, and
consequently to be contaminated by bacteria. The presence of high-cross linking structure
of the membranes influences their stiffness and also the necessary time to full reabsorb the
material: higher is this parameters, higher is the risk to expose in the oral cavity, to
contaminate with bacteria, and to induce a clinical failure. So, the ideal material, should
promote the cellular interaction, proliferation, and vascularization, in order to increase the
healing process and being reabsorbed in a shorter period [Barbeck M, Lorenz J, Kubesch
A, Böhm N, Booms P, Choukroun J, Sader R, Kirkpatrick CJ, Ghanaati S. Porcine DermisDerived Collagen Membranes Induce Implantation Bed Vascularization Via Multinucleated
Giant Cells: A Physiological Reaction? J Oral Implantol. 2015 Dec;41(6):e238-51. doi:
10.1563/aaid-joi-D-14-00274. Epub 2014 Dec 29. PMID: 25546240].
2. The authors should remove the word novel from the title because this formulation is not
novel.
AUTHOR’S ANSWER: Thank you for your request. We have removed the word novel from
the title.
3. Are there any notable differences were observed from SEM images between the two
tested membranes?
AUTHOR’S ANSWER: Thank you for your comment. It is important to specify that we have
analyzed the same membrane in two conditions: with and without hydration. In the Fig. 1 we
have observed at SEM the membrane without cells. Although in both conditions it is showed
a smoothed membrane, H-membrane appeared more porous than manufactured membrane
(Membrane). In the Fig. 4-5 we have observed gingival fibroblasts and osteoblasts cultured
on the tested matrices. Since cells colonized the entire surface producing a dense layer of
cells, the surface of the matrix cannot be observed.
4. As the authors mentioned, the specific aim of the study includes the investigation of the
topographical and biological properties of the porcine acellular dermal matrix. It would be
better if authors can explain some of the possible surface/topological factors (such as
membrane roughness) that could cause the greater proliferation of HOB cells than the HBF
cells.
AUTHOR’S ANSWER: Thank you for your comment. We cannot consider the proliferation
of HOBs highest than HGFs, since the difference was not statistically significant. We have
also observed that the matrix promoted cell adhesion similarly . Furthermore, these
mechanisms were influenced from the activities of Cadherins, that favored and influenced
the adhesion of fibroblast and osteoblast [Chan MW, El Sayegh TY, Arora PD, Laschinger
CA, Overall CM, Morrison C, McCulloch CA. Regulation of intercellular adhesion strength in
fibroblasts. J Biol Chem. 2004 Sep 24;279(39):41047-57. doi: 10.1074/jbc.M406631200.
Epub 2004 Jul 6. PMID: 15247242.]. In addition, the mebrande in both conditions (with and
without hydration) is characterized by a dense network of collagen fibers, that favored a
more easily adhesion and attachable of these types of cells through the cell matrix
[Shekaran A, García AJ. Extracellular matrix-mimetic adhesive biomaterials for bone repair.
J Biomed Mater Res A. 2011 Jan;96(1):261-72. doi: 10.1002/jbm.a.32979. Epub 2010 Nov
10. PMID: 21105174; PMCID: PMC3059117; Nishimura K. Effect of extracellular matrix and
serum components on cellular adhesion and growth in vitro and in vivo. Nichidai Koko
Kagaku. 1990 Jun;16(2):237-60. Japanese. PMID: 2135612].
We have introduced these concepts in the discussion: “One potential explanation for this
may be because both membranes possess a dense network of collagen fibers, thereby
providing a more easily attachable surface for in-coming HGFs and HOBs. In addition, the
adhesion could be promoted from the Cadherins activities”.
5. Could authors explain the purpose of using different magnifications for H-membrane (A,
C) and Membrane (B, D) in Fig. 2?
AUTHOR’S ANSWER: Thank you for your observation. There was a mistake in the caption,
all the figures were at the magnification of 40x. We have corrected it.
6. The authors have not explained why the proliferation rate in Fig. 3 is inconsistent on
different days. The authors should have addressed it appropriately.
AUTHOR’S ANSWER: Thank you for your observation. We observed an intermittent
proliferation rate in HGF. In detail, we observed an increased of cell growth at 3 days that
decreased at 7 days and finally augmented again. This trend is characteristic of fibroblasts
as demonstrated in literature [Espósito ACC, Brianezi G, Miot LDB, Miot HA. Fibroblast
morphology, growth rate and gene expression in facial melasma. An Bras Dermatol. 2022
Sep-Oct;97(5):575-582. doi: 10.1016/j.abd.2021.09.012. Epub 2022 Jul 12. PMID:
35840442; PMCID: PMC9453522]. On the contrary osteoblasts have a different growth rate,
their proliferation augmented in time dependent manner [Zhou J, Li B, Lu S, Zhang L, Han
Y. Regulation of osteoblast proliferation and differentiation by interrod spacing of Sr-HA
nanorods on microporous titania coatings. ACS Appl Mater Interfaces. 2013 Jun
12;5(11):5358-65. doi: 10.1021/am401339n. Epub 2013 May 21. PMID: 23668394.]
We have added these aspects in the discussion: “In detail, for HGFs a statistically increase
of cell growth (p<0.0001) was observed at 3 and 14 days, while at 7 days the cellular
metabolic activity was reduced. This trend was characteristics for fibroblasts”.
7. We have found that some of the SEM images were duplicated in Fig. 4. For example,
Figs. 4G and J, and Figs. 4 M and O, looks the same at different magnifications. It looks like
Felice et al. has used single SEM images multiple times, which is not good for the scientific
community.
AUTHOR’S ANSWER: Thank you very much for your observation. We are agreed with you.
There was a mistake, thus we have inserted the right figures. In addition, thanks to your
comment we have seen a further mistake with the magnification. Therefore, we have
provided to correct all the figure.
Minor:
8. Normalize the materials and methods section 4.1.
AUTHOR’S ANSWER: Thank you for your request. We have normalized this paragraph.
9. The caption for Figs. 2 and 3 are inconsistent. The team should use a uniform format for
figure captions and figure citations.
AUTHOR’S ANSWER: Thank you for your comment. We have uniformed the Fig. 2 with the
others, and we have modified the caption.
10. Fig. 5 – ‘days’ were missing in the Figure caption.
AUTHOR’S ANSWER: Thank you for your observation. We have added the missing word.
11. The authors should have used the proper formatting options. It is a bit surprising to see
different font styles in the middle of the manuscript.
AUTHOR’S ANSWER: Thank you for your comment. We have uniformed the style of the
manuscript.
Similarly, the English language needs tobe improved and there is some inconsistency in
writing and also has a lot of grammatical issues;
a. Typo: Page 1, line 32 – mainly;
b. Page 2, line 52, 55, 58;
c. Page 3, Fig 2. labels:
d. Page 9, lines 295-301;
e. Figure 9, 10 captions;
AUTHOR’S ANSWER: Thank you for your observations. We have checked the English in
the entire manuscript.
Round 2
Reviewer 3 Report
Figure 1: Labels were mismatched. I think (A-D) corresponds to Mag 300x, not (A-B). Similarly, correct the others as well.
Figure 4: ‘Days’ were missing.
Author Response
Figure 1: Labels were mismatched. I think (A-D) corresponds to Mag 300x, not (A-B). Similarly, correct the others as well.
Figure 4: ‘Days’ were missing.
AUTHOR'ANSWER: thank you very much for your comments; we have corrected the figures.
The English has been checked and corrected.